# Exploring a Possible Interplay between Schizophrenia, Oxytocin, and Estrogens: A Narrative Review

**DOI:** 10.3390/brainsci13030461

**Published:** 2023-03-08

**Authors:** Danae Papadea, Christina Dalla, Despina A. Tata

**Affiliations:** 1Laboratory of Cognitive Neuroscience, School of Psychology, Aristotle University of Thessaloniki, 54124 Thessaloniki, Greece; 2Department of Pharmacology, Medical School, National and Kapodistrian University of Athens, 11743 Athens, Greece

**Keywords:** estrogen hypothesis, oxytocin, psychosis, schizophrenia, sex hormones

## Abstract

Schizophrenia is characterized by symptoms of psychosis and sociocognitive deficits. Considering oxytocin’s antipsychotic and prosocial properties, numerous clinical, and preclinical studies have explored the neuropeptide’s therapeutic efficacy. Sex differences in the clinical course of schizophrenia, as well as in oxytocin-mediated behaviors, indicate the involvement of gonadal steroid hormones. The current narrative review aimed to explore empirical evidence on the interplay between schizophrenia psychopathology and oxytocin’s therapeutic potential in consideration of female gonadal steroid interactions, with a focus on estrogens. The review was conducted using the PubMed and PsychINFO databases and conforms to the Scale for the Assessment of Narrative Review Articles (SANRA) guidelines. The results suggest a potential synergistic effect of the combined antipsychotic effect of oxytocin and neuroprotective effect of estrogen on schizophrenia. Consideration of typical menstrual cycle-related hormonal changes is warranted and further research is needed to confirm this assumption.

## 1. Introduction

Schizophrenia is a neurodevelopmental mental disorder associated with positive and negative symptoms, as well as sociocognitive deficits [1]. Unravelling schizophrenia’s complexity and heterogeneity is challenging and knowledge on the underlying etiological mechanisms remains scarce. Due to rapid progress in neuroscience and genetic research, a neurobiological approach has been adopted. This approach identifies schizophrenia as a series of neurodevelopmental, neuropathological, and brain plasticity abnormalities (e.g., [2,3]). Treatment is achieved via antipsychotic medication, accompanied by psychotherapy, although psychotherapeutic intervention and medication can only reduce the symptoms of the disorder [4].

In the early 70′s, a neuropeptide hormone principally known for its role in parturition and lactation, oxytocin (OXT), gained attraction as a potential treatment agent for schizophrenia and general psychopathology [5], based on its antipsychotic and prosocial properties in clinical and preclinical studies. Oxytocin’s synthesis primarily occurs in magnocellular neurons of the paraventricular (PVN) and supraoptic (SON) nuclei in the hypothalamus [6]. Following synthesis, OXT is transferred to the posterior lobe of the pituitary and is released directly into the bloodstream in response to various somatosensory stimuli, acting as a hormone in the peripheral nervous system [7]. Meanwhile, OXT acts as a neurotransmitter and neuromodulator by interacting with the central OXT receptor (OXTR) in various brain sites, such as the nucleus accumbens and the hippocampus [6,8,9].

Oxytocin’s central and peripheral role has been extensively examined in numerous physiological and psychological studies. Other than its contribution to maternal care and infant–mother bond formation [10], sexual behavior, and reproduction [11], OXT enhances prosocial behavior by reinforcing empathy [12], cohesion, and co-operation [13], as well as social memory [14], and emotional perception [15]. In addition to social behavior, OXT is proposed as a regulator of anxiety, with anxiolytic and anxiogenic properties [16]. Consequently, OXT is considered a regulator of human behavior, which indicates that the oxytocinergic system may be implicated in schizophrenia symptomatology [17]. Meanwhile, studies support the hypothesis that exogenous OXT treatment reduces symptoms of schizophrenia (e.g., [18,19]), findings that have encouraged researchers to further explore OXT’s therapeutic efficacy.

Sex modulates the clinical course of schizophrenia (e.g., later age of psychosis onset in female compared to male patients) [20]. Oxytocin also regulates social behavior in a sex-specific manner, namely different OXT-related behavioral patterns appear in male and female populations [21]. For instance, OXT facilitates a tend-and-befriend behavior [22], social approach [23], altruism [24], and positive social judgement in the female population, whereas in males, negative social judgement [25], competition [26], and social avoidance [27] are fostered. These sex-associated differences indicate the possible involvement of gonadal steroid hormones (i.e., androgens, estrogens, and progestogens) [28,29]. Indeed, endogenous OXT concentration follows a pattern similar to estrogens concentration in an average 28-day menstrual cycle [30]. Meanwhile, in patients diagnosed with schizophrenia, a peak in estrogens correlates with reduced symptom severity, known as the estrogen hypothesis [31].

The aim of the present narrative review is to explore an interplay between the psychopathology of schizophrenia and the therapeutic potential of OXT, taking into consideration interactions with gonadal steroids, such as estrogens. To date, no clinical trial has investigated the prospect of exogenous OXT administration (monotherapy, or adjacent to antipsychotic medication) while controlling for typical menstrual cycle-related hormonal changes. To the best of our knowledge, this is the first review to discuss the possibility of an interaction between OXT’s antipsychotic and estrogens’ neuroprotective effects in schizophrenia. A synergistic action of combined estrogens and OXT on the clinical course of schizophrenia is hypothesized.

## 2. Methods

This narrative review conforms to the Scale for the Assessment of Narrative Review Articles (SANRA) guidelines [32]. The databases PubMed/Medline and PsycINFO were searched for articles containing the main keywords (“schizophrenia” OR “psychosis” OR “antipsychotics”) AND (“oxytocin”) OR (“estrogens” OR “gonadal hormones” OR “sex differences” OR “menstrual cycle”), published in English, between January 1990 and February 2023. Secondary search engines (e.g., Google Scholar) were also utilized and an extensive search of the sourced articles revealed additional references. A total of 27,335 articles were screened based on their significance and relevance of their information (N = 120). Clinical trials and preclinical studies, observational or case-control studies, prospective cohort studies, as well as reviews and meta-analyses were included. No exclusions were performed based on participant sex or age.

## 3. The Oxytocin System in Schizophrenia

Several clinical and preclinical studies provide salient evidence in support of OXT’s natural antipsychotic properties (e.g., [19,33]). In an attempt to explore OXT’s effect on schizophrenic behavior, research focuses on the exogenous OXT’s therapeutic efficacy in alleviating positive and reducing negative symptoms and sociocognitive deficits. However, research on the effects of either endogenous OXT or exogenous administration of the neuropeptide has led to controversial results and the antipsychotic-like effect of the neuropeptide is still under consideration [34].

### 3.1. The Endogenous Oxytocin System in Schizophrenia

Studies indicate that the endogenous OXT system is perturbed in individuals diagnosed with schizophrenia compared to healthy individuals. These abnormalities correspond to either increased or decreased levels of OXT in relation to normal values. For instance, Walss-Bass and colleagues found increased OXT in the plasma of schizophrenic patients with delusional symptoms, suggesting that the presence of delusions may drive secretion of OXT [35]. On the contrary, Jobst and colleagues reported significantly lower plasma OXT levels in schizophrenia speculating that the decrease may be due to altered OXT metabolism or a decrease in OXT synthesis and in mRNA expression and translation [36]. Similarly, significantly lower OXT plasma levels have been reported in first-episode schizophrenia patients, although mRNA expression of OXT and OXTR genes were higher than in healthy controls [37].

According to recent systematic reviews and meta-analyses, endogenous OXT concentrations are typically lower in the plasma but higher in cerebrospinal fluid (CSF) of schizophrenic populations [38,39]. Low plasma OXT levels are associated with increased severity, impaired functionality, and a worse clinical outcome of the disorder [38]. High OXT levels in the CSF may be explained by coregulation between the hypothalamic–pituitary–adrenal (HPA) and OXT system [39]. Specifically, OXT inhibits activity of the HPA axis by modulating production and secretion of cortisol [40]. However, in response to stress, perturbation of the suprachiasmatic nucleus’ function may also result in altered endogenous release of OXT in the central and peripheral nervous system [41]. The OXT system appears to be dysregulated in the pathophysiology of schizophrenia, but the exact endogenous profile is yet to be elucidated.

#### Positive and Negative Symptoms

After OXT’s natural antipsychotic properties were introduced, the relationship between the neuropeptide and positive symptoms of schizophrenia attracted significant attention. According to Rubin and colleagues, higher levels of OXT in plasma correlate with a reduction in positive symptoms in female patients with chronic schizophrenia [42]. In addition, recent studies support a positive association between plasma OXT and functionality in schizophrenia, whereas lower endogenous OXT levels have been linked to acute positive symptomatology (e.g., [43,44]). However, Rubin and colleagues have also noted that high levels of OXT in plasma may also be associated with increasing positive symptoms [45]. According to the authors, this discrepancy may be attributed to a greater severity in psychotic symptoms, compared to the symptoms experienced by the sample recruited in the earlier study [42,45]. In addition, OXT plasma levels were also higher, hence, OXT may include benefits when endogenous OXT levels are low, but display unfavorable outcomes at a higher concentration [45].

Considerable research has been devoted to the association between OXT levels and negative symptoms, including cognitive and sociocognitive deficits. Generally, in schizophrenic patients, higher plasma OXT is associated with improved social cognition, processing speed, working memory, affective theory of mind (ToM), and facial emotion recognition [46,47,48,49]. In contrast, lower plasma OXT relates to greater asociality, poor metacognition, low facial emotion recognition accuracy, and may predict emotional and social withdrawal in schizophrenia [50,51,52,53]. For instance, patients with schizophrenia exhibited lower levels of OXT after trust-related interactions compared to healthy controls, namely when participants were asked to share an important secret with the experimenter, an action requiring trustworthiness [51]. Moreover, an increase in endogenous OXT levels predicts accurate encoding of lower-level socially relevant information, namely subtle social cues, such as facial expressions [54,55,56]. In addition, sex differences indicate that higher endogenous OXT levels correlate with greater performance in the emotional identification of body gestures in female, compared to male subjects, while female patients with schizophrenia perceive facial expressions as happier when OXT levels are high [47,55]. Relative sociocognitive deficits can be further explained by downregulation of OXTR mRNA in brain regions involved in social cognition, such as the temporal cortex [57].

### 3.2. Exogenous Oxytocin Effects in Schizophrenia

#### 3.2.1. Clinical Studies

Clinical studies typically utilize intranasal OXT treatment in patients diagnosed with schizophrenia who are stable on antipsychotic medication (Table 1). In general, OXT administration seems to improve both negative and positive symptoms in numerous studies. Specifically, daily intranasal administration of OXT in the span of 3 to 12 weeks has shown a significant improvement in positive and negative symptoms, as well as general psychopathology [19,58,59,60,61]. Likewise, exogenous sublingual OXT administration, in addition to clozapine treatment, reduced negative symptoms and maintained low positive symptoms in young adult patients with treatment-resistant schizophrenia [62]. Meanwhile, a single dose of intranasal OXT increased functional connectivity between the amygdala and left middle temporal gyrus (MTG), superior temporal sulcus (STS), and angular gyrus (AngG), associated with a reduction in negative symptoms, as well as the caudate and left supplementary motor area, precentral gyrus, and frontal inferior triangular gyrus, associated with greater cognitive insight and lower negative symptoms [63,64].

Exogenous OXT treatment has been also associated with improvements in the sociocognitive deficits of schizophrenia. For instance, participants showed improvement in the ability to recognize and identify emotions, as well as indirectly expressed emotions, thoughts, and intentions, following 20–40 IU intranasally administered OXT [65,66,67]. Similarly, acute OXT treatment enhanced emotion recognition accuracy in a task that required social cue processing [68]. According to a neurofunctional social cognitive model proposed by Rosenfeld and colleagues, improper oxytocinergic and dopaminergic signals in the amygdala lead to impaired emotional salience processing that engenders social cognitive deficits, as observed in schizophrenia [82]. For instance, exogenous intranasal administration of OXT (40 IU) enhanced amygdala reactivity in response to emotional faces in healthy controls, whereas in schizophrenic patients, the same OXT intranasal dose attenuated amygdala reactivity during the processing of emotional faces [69]. A recent functional magnetic resonance imaging (fMRI) study further supports reduced activation in the amygdala, among other brain regions (i.e., temporo-parietal junction, posterior cingulate cortex, precuneus, and insula) which are involved in the processing of facial emotion, salience, aversion, uncertainty, and ambiguity in social stimuli [70]. In fact, intranasal OXT administration reduces neural activity in patients with schizophrenia or schizoaffective disorder, within the aforementioned regions when presented with happy and angry faces [70].

Additionally, longitudinal OXT intranasal administration has led to a significant improvement in the perception of trustworthiness and ToM, interpersonal empathy, as well as verbal memory and learning, in male and female participants with schizophrenia [71,72,73]. Administration of a single dose of OXT has led to similar results in the improvement of ToM, i.e., perception of indirect hints and social faux pas, indirect emotions, thoughts, and intentions, and higher-level social cognition, i.e., sarcasm, deception, empathy [18,67,74]. In fact, empathy seems to increase after both 40 IU and 24 IU of OXT in patients with schizophrenia [75,76].

However, several randomized controlled clinical studies have failed to replicate a therapeutic effect for positive and negative symptomatology. For instance, 6 weeks of intranasal OXT treatment in patients with schizophrenia or schizoaffective disorder and individuals with early psychosis showed no evidence of OXT’s therapeutic efficacy on negative symptoms, cognitive deficits, and social behavior, functioning, and cognition [73,77,78,79]. Similarly, OXT did not improve facial emotion processing and recognition in patients diagnosed with schizophrenia, following 40–48 IU of intranasal OXT [18,80]. Meanwhile, acute OXT treatment did not improve mentalizing (i.e., the ability to infer other people’s intentions and emotions) in female patients with schizophrenia, whereas the opposite effect has been shown in males, which indicates sex-specific differences in OXT treatment [81]. Discrepancy in the literature may be attributed to various factors. In addition to dosage differences, differences in sample sizes, age, and symptom severity, as well as time between OXT administration and behavioral testing should be considered (e.g., [18,19]). After administration, a 50 min duration is estimated for the neuropeptide to take effect, which might explain the lack of OXT’s therapeutic effect on sociocognitive deficits after 15–30 min of administration [26,83]. Additionally, recent meta-analyses argue that OXT’s efficacy is evident at higher doses (>40–80 IU) [84,85].

#### 3.2.2. Preclinical Studies

Numerous researchers have introduced a therapeutic-like effect of exogenous OXT in various preclinical studies (Table 2). For example, Feifel and Reza were the first to demonstrate this therapeutic effect of OXT on an animal model of schizophrenia [86]. Their study focused on identifying how OXT regulates impairments in prepulse inhibition (PPI), a measure of sensorimotor gating found to be deficient in schizophrenia. Administered OXT was able to restore impaired PPI in rats [86]. Similarly, OXT attenuated PPI deficits in inbred high-avoidance rats and increased PPI in outbred heterogeneous stock rats, supporting OXT’s anti-psychotic treatment potential [87].

Sociability may also be enhanced by acute OXT treatment. For example, OXT-treated rats exhibit greater levels of social interaction (e.g., time in proximity) compared to vehicle-injected rats [88,89]. Similarly, rhesus macaques were more likely to choose to reward another monkey, when the alternative was to reward no monkey at all, after OXT treatment [90]. On the contrary, chronic OXT administration has not improved social interaction based on preclinical data. For instance, repeated OXT treatment led to aggressive behavior, greater cortisol levels, and reduced time in social interaction among neonatal pigs [91]. Similarly, consecutive OXT treatment decreased social interaction with the opposite, as well as the same, sex in mice and impaired partner preference behavior (e.g., contact with opposite-sex partner) in prairie voles [92,93]. Consequently, it can be assumed that prolonged exposure to OXT treatment diminishes sociability in animals without laboratory induced sociocognitive deficits, whereas acute OXT treatment increases social interaction [92]. Therefore, the duration of OXT administration and treatment should be taken into consideration.

Meanwhile, social recognition, social and spatial memory and learning, and generally cognitive performance seem to improve after both acute and long-term OXT treatment in animal models. Social perception was enhanced after intranasal OXT administration in rhesus macaques by reducing attention to negative stimuli [94]. In addition, Ferguson and colleagues demonstrated social amnesia in OXT knockout mice, but OXT intracerebroventricular injections were able to restore social memory, whereas treatment with an OXT-antagonist induced social-amnesia in mice with an intact OXT system [14].

## 4. Involvement of Sex Hormones

Studies exploring OXT’s neural and behavioral activity suggest differences between male and female participants. Although OXT promotes social behavior in females, OXT seems to facilitate negative social approach in males (e.g., [25]). Additionally, peripheral concentrations of OXT modulate resting brain activity differently in male and female patients with schizophrenia (e.g., low OXT levels are associated with lower amplitude of low-frequency fluctuations in the frontal cortex in females, but in posterior cingulate in males) [95]. These differences may be explained by interaction of the OXT system with gonadal hormones. For instance, testosterone, the main androgens hormone, promotes OXT binding in the hypothalamus, while exogenous OXT increases plasma testosterone in males [96,97]. Moreover, estrogens appear to regulate the OXT system by altering OXT and OXTR mRNA levels in the brain, due to modulation of the OXTR gene expression [98,99]. In fact, 17β-estradiol (E2), the most potent estrogen, stimulates OXT synthesis and dendrosomatic release [100,101]. Interactions between OXT and estrogens have been proposed in both preclinical and clinical studies (e.g., [102,103]). Meanwhile, progesterone inhibits OXT binding; although, OXT accelerates follicle-stimulating hormone (FSH)- and forskolin (FSK)-induced progesterone production [104,105].

It is evident that the major sex hormones, namely testosterone, estrogens, and progesterone, interact with the endogenous OXT system. However, cyclic hormonal changes are typical during the menstrual cycle (Figure 1) [106]. Thus, it is essential to consider any hormonal variations when evaluating the neuropeptide’s efficacy in patients with schizophrenia.

Oxytocin is able to modulate the menstrual cycle and ovulatory phase by affecting follicular luteinization and ovarian steroidogenesis [108]. Plasma OXT concentration increases during the follicular to ovulatory phase, whereas a decrease is observed from ovulation to the midluteal phase [30,108]. According to Engel and colleagues, estradiol concentrations peak in women before ovulation; hence, there is an increase in estradiol availability and estrogen receptor beta functioning [30]. During the same phase of the menstrual cycle, there is also a peak in OXT, which could be associated with the increase in estradiol availability (Figure 2) [30]. Indeed, plasma OXT levels positively correlate with estrogens levels, but correlate negatively with progesterone levels [109].

### 4.1. Estrogen Hypothesis

Early in the 19th century, clinicians formed the so-called *estrogen hypothesis*, after observations of a possible relationship between the menstrual cycle and psychotic symptoms in schizophrenic female patients [31]. According to the estrogen hypothesis, which was derived from numerous clinical studies, estrogens act protectively against the development and severity of schizophrenia [110,111]. For instance, earlier menarche, which stimulates the production of estrogens, among other steroid hormones, is associated with later onset of schizophrenia, while post-menopause, associated with a decline in estrogen levels, is characterized by an exacerbation of psychosis symptoms [110,112].

Estrogen concentration rises during the midfollicular phase and decreases post-ovulation, followed by another slight increase in the midluteal phase [106]. Overall, during the high estrogen phase of the menstrual cycle (ovulatory and midluteal) negative and global symptom scores have been reported, while positive symptom improvement is observed during the progesterone phase (i.e., luteal phase) [113,114,115]. In contrast, the risk of psychiatric admission is typically higher during the perimenstrual phase, when estrogen levels are low [113,116,117].

Estrogen augmentation has been proposed as an effective treatment strategy in improving total symptom severity and reducing positive and negative symptoms in female patients diagnosed with schizophrenia [118,119,120]. However, estrogen augmentation may exhibit significant side effects in long-term use in both sexes (e.g., feminizing effects in males); therefore, selective estrogen receptor modulators (SERMs) have been proposed as a safer alternative (e.g., raloxifene) [118,121,122].

#### Antipsychotics and Estrogens

Differences in the efficacy of antipsychotic medication among male and female patients diagnosed with schizophrenia spectrum disorders have been reported [123]. The prevailing form of estrogen, E2, increases dopamine (D_2_) receptor sensitivity in the ventral tegmental area; therefore, higher D_2_ receptor occupancy of antipsychotic drugs is observed in female patients [124,125]. In addition, drug absorption and bioavailability is higher, whereas drug metabolism is lower in the female population [125]. In turn, concentration-to-dose ratios are significantly higher in women than men for antipsychotics (e.g., clozapine, haloperidol, olanzapine, and risperidone) [126]. Therefore, considering estrogen fluctuations throughout the menstrual cycle, dose deductions of antipsychotics during the midluteal and ovulatory phase and dose increments during the perimenstrual phase have been recommended [125].

In addition, chronic use of antipsychotics (e.g., risperidone) increases the risk for hyperprolactinemia, which results from D_2_ receptor blockage in the anterior pituitary gland and enhanced prolactin secretion [112,127,128]. In turn, prolactin secretion results in estrogen deficiency and dopamine stimulation, further exacerbating schizophrenia symptoms [120]. Therefore, prolactin-sparing antipsychotics are recommended for female patients with schizophrenia [125]. Therefore, considering efficacy and acceptability, quetiapine is usually preferred among prolactin-sparing antipsychotics [129].

## 5. Schizophrenia, Oxytocin, Estrogens: A Possible Interplay

The purpose of this narrative review was to accentuate an interplay between schizophrenia psychopathology, OXT’s therapeutic potential, and estrogen’s neuroprotective effect. Indeed, the endogenous OXT system appears to be dysregulated in the clinical course of schizophrenia, while exogenous OXT is proposed as an antipsychotic and prosocial agent [34,39]. Sex-associated differences in both schizophrenia and the OXT peripheral system suggest the involvement of gonadal steroid hormones (e.g., [95,110]). Moreover, exogenous OXT treatment may have sex-specific effects in certain behavioral outcomes (e.g., mentalizing improvement in male, but not in female, participants following OXT administration) [81]. Thus, sex hormonal fluctuations during the female menstrual cycle are an essential consideration in the psychopathology of schizophrenia and in the dysregulation of the OXT system. The ovulatory and midluteal phase of the menstrual cycle, characterized by high estrogen levels, positively correlate with higher OXT levels in plasma and improvement in psychotic symptoms (the estrogen hypothesis) [30,109,110].

Considering these independent interactions, Rubin and colleagues suggested that symptoms of psychosis vary across the menstrual cycle in female patients diagnosed with chronic schizophrenia [42]. Symptom severity decreased during the midluteal phase when estrogen levels were high, a finding that complies with the estrogen hypothesis [42,110,130]. Moreover, endogenous OXT levels, on average, were associated with an improvement in positive symptoms, social behavior, and general psychopathology, although plasma OXT did not significantly fluctuate across cycle phases in the schizophrenic population [42]. In nonschizophrenic female participants, OXT levels tend to vary throughout the menstrual cycle, namely lower OXT levels during the perimenstrual phase [30,109]. The discrepancy in OXT concentrations may be due to dysregulation of the endogenous OXT system in schizophrenia [39].

A possible explanation for the dysregulation of the OXT system may be attributed to antipsychotic- and stress-induced hyperprolactinemia, resulting in estrogen deficiency [127]. Indeed, preclinical data note that prolactin inhibits the secretion of OXT in nonlactating virgin rats [131]. Given that E2 stimulates OXT synthesis and release [101], prolactin-sparing antipsychotics that act protectively against estrogens deficiency [120] may act against OXT dysregulation in schizophrenia as well. Meanwhile, OXT influences ovulation by affecting follicle luteinization and ovarian steroidogenesis and promoting estrogen secretion [108]. Thus, exogenous OXT treatment and estrogen augmentation pose promising therapeutic benefits in schizophrenia (e.g., [63,120]) and their combination may act synergistically. Considering that psychosis symptoms are often exacerbated during the perimenstrual phase [113], antipsychotic (prolactin-sparing, e.g., quetiapine) augmentation with SERMs (e.g., raloxifene) and/or intranasally administered OXT may act as a better alternative than incremented antipsychotic dosage in treatment-resistant schizophrenia.

Current empirical evidence suggests an interaction between the OXT system and estrogen, translating into similar concentration patterns across the female menstrual cycle (e.g., [109]). Improvement in schizophrenia psychopathology, associated with higher plasma OXT levels, is observed during the midluteal phase, when estrogens levels are high [42]. Considering the independent potential of OXT and estrogens in ameliorating schizophrenia symptoms (e.g., [71,130]), a synergistic effect of the combined antipsychotic effect of exogenous OXT and neuroprotective effect of estrogen on the improvement of schizophrenia symptoms may be a novel, but a rational assumption.

## 6. Limitations and Future Directions

Although the present review was able to provide a comprehensive overview of the independent interactions between schizophrenia, OXT, and sex hormones, there are certain limitations. First, the interplay was proposed under the assumption that OXT is a neuropeptide with prosocial and antipsychotic properties [132]. However, according to the social salience hypothesis, OXT does not enhance prosocial behavior, but simply regulates attention to social cues [133]. This framework could explain the differences between the effects of acute versus long-term OXT treatment in sociability (e.g., [92]), as well as the general discrepancy in results within the literature of schizophrenia (e.g., [42,45]). Additionally, OXT is administered supplementary to other treatments (e.g., clozapine) and a randomized controlled clinical trial utilizing an OXT monotherapy model is still lacking [134]. Although animal models may fill this gap, a clinical monotherapy treatment trial is proposed to evaluate OXT’s effects in schizophrenia, controlling for symptom severity, trial duration, and participant age.

The validity of measuring plasma OXT levels in the peripheral system as an indicator of OXT levels in the central nervous system is also debated, most likely due to the restricted permeability of neuropeptides across the blood–brain barrier (BBB) [135]. The endogenous OXT system appears to be dysregulated in schizophrenia (i.e., higher in CSF, lower in plasma), but CSF OXT concentration is considered a better proxy for central activity than plasma levels [136]. Therefore, plasma OXT concentration may provide minimal information of the neuropeptide’s central activity in schizophrenia psychopathology. Exogenous OXT administration can reach both the CSF in the brain ventricular system and circulatory system, but careful interpretation of findings in plasma concentrations is needed [136]. We propose that OXT in CSF could be utilized as a measure of the neuropeptide activity in the central nervous system instead.

Another limitation is the inclusion of empirical data, which are not sex-specific. Although our focus was on estrogen concentration, we explored the interaction between the OXT system and schizophrenia in studies that included and analyzed male and female participants altogether. However, the relationship between the neuropeptide and schizophrenia may be susceptible to sex-specific effects of OXT’s neural and behavioral activity (e.g., [137]). According to recent recommendations, it should also be investigated whether OXT’s therapeutic potential is sex-differentiated [138]. Moreover, the endogenous OXT system in schizophrenia appears to be dysregulated but it is not yet clear whether it translates into incremented or reduced levels of the neuropeptide in the central or peripheral system [30,42]. The endogenous OXT profile in female patients diagnosed with schizophrenia and healthy individuals may be dissimilarly influenced by various factors, such as menstrual cycle irregularities, contraceptive use, antipsychotic medication, etc., (e.g., [139]). Thus, the interactions between OXT and estrogen, as well as OXT and schizophrenia, need to be further clarified. We propose that future studies should investigate the endogenous OXT profile in female patients diagnosed with schizophrenia, with attention to hormonal changes during a typical menstrual cycle.

## 7. Conclusions

Independent correlations between schizophrenia, OXT, and estrogens are evident, i.e., schizophrenia–OXT, estrogen–OXT, schizophrenia–estrogen. Therefore, the interaction between estrogen’s neuroprotective features and OXT’s antipsychotic and prosocial properties may have a synergistic therapeutic effect in schizophrenia’s psychopathology. To support this hypothesis, further empirical evidence in clinical and preclinical studies is warranted. Furthermore, menstrual cycle-related hormonal fluctuations are an essential consideration in psychopharmacological treatment and the clinical course of psychopathology, in general.

## Figures and Tables

**Figure 1 brainsci-13-00461-f001:**
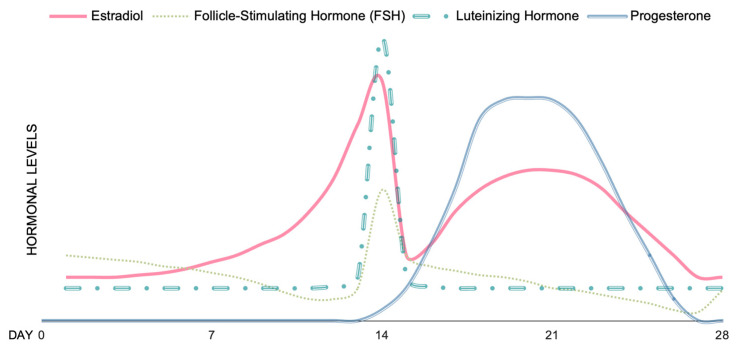
Estimated hormonal levels during an average healthy menstrual cycle of 28 days (based on data from [106,107]).

**Figure 2 brainsci-13-00461-f002:**
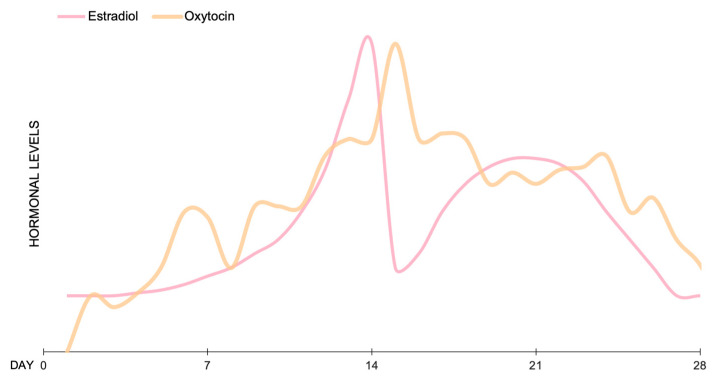
Estimated hormonal levels of OXT and estradiol during an average menstrual cycle of 28 days (based on data from [30,106,107]).

**Table 1 brainsci-13-00461-t001:** Clinical Studies.

Author, Year	Design	Subjects N (M/F)	Trial Duration	OXT Administration	Testing after Dose	Outcome Measures	Results (OXT)	Conclusions *
Davis et al., 2013 [18]	RDBPC	23 (23M) SCHZ	3 sessions	40 IU or placebo; intranasal	30 min (3rd session)	Social cognition	Improved inferential and regulatory processes.	Acute therapeutic effects on higher-order social cognition.
Feifel et al., 2010 [19]	RDBPC/CO	15 (12M, 3F) SCHZ	7 weeks	2 × 20 IU daily for 1 week, 2 × 40 IU thereafter or placebo; intranasal	N/A	Clinical symptoms, symptom severity	Clinical symptom improvement at 3-weeks.	Antipsychotic properties, with a delayed onset of action.
Gibson et al., 2014 [58]	RDBPC	14 (11M, 3F) SCHZ	6 weeks	2 × 24 IU or placebo; intranasal	50 min after the AM dose at the end of week 6	Clinical symptoms, social cognition, and skills	Improved negative symptoms, fear recognition, self-reported perspective taking.	Limited efficacy on social cognition and social skills.
Jarskog et al., 2017 [59]	RDBPC	62 (47M, 15F) SCHZ or schizoaffective disorder	12 weeks	2 × 24 IU or placebo; intranasal	N/A	Clinical symptoms, social cognition and skills	Improved social skills and negative symptoms.	No significant effect on sociocognitive function.
Modabbernia et al., 2013 [60]	RDBPC	40 (33M, 7F) SCHZ, stable on risperidone	8 weeks	2 × 20 IU daily for 1 week, 2 × 40 thereafter or placebo; intranasal	N/A	Clinical symptoms	Clinical symptom improvement at 8-week end point.	Therapeutic effect additive to risperidone treatment.
Ota et al., 2018 [61]	NROL	16 (7M, 9F) chronic SCHZ	12 weeks	2 × 12 IU daily; intranasal	N/A	Clinical symptoms, cognition, social cognition, brain structure (MRI)	Clinical symptom improvement, negative correlation with gray matter of right insula and left cingulate cortex, verbal fluency improvement.	Therapeutic effect associated with gray matter volume changes.
Marotta et al., 2020 [62]	Retrospective Medical Chart Review	5 (4M, 1F) treatment resistant SCHZ	1 year	10 IU–3 × 20 IU daily; sublingual (+ clozapine, 200–600 mg)	N/A	Clinical symptoms, social functioning	Reduced negative, maintained low positive symptoms, and improved occupational and social functioning.	Augmentation of clozapine with OXT effective in reducing negative symptoms.
Abram et al., 2020 [63]	RDBPC/CO	22 (22M) SCHZ, 24 (24M) HC	2 sessions, 2 weeks apart	40 IU or placebo; intranasal	90 min	Functional connectivity	Enhanced functional connectivity between amygdala and MTG/STS/AngG	OXT can normalize amygdala circuit associated with negative symptoms.
Korann et al., 2022 [64]	PC	31 (31M) SCHZ, 21 (21M) HC	2 sessions	24 IU or placebo; intranasal	45 min	Functional connectivity	Enhanced connectivity between left caudate, left supplementary motor area, left precentral gyrus, left frontal triangular gyrus	Enhanced connectivity in regions associated with cognitive insight and negative symptoms.
Averbeck et al., 2012 [65]	RDBPC/CO	21 (21M) SCHZ	2 sessions, approx. 7–8 days apart	24 IU or saline placebo; intranasal	50 min	Emotion recognition	Improved recognition of basic emotions.	Acute therapeutic effect on emotion recognition.
Goldman et al., 2011 [66]	RDBPC	5 (3M, 2F) polydipsic SCHZ, 8 (4M, 4F) non-polydipsic SCHZ, 11 (4M, 7F) HC	3 sessions, approx. 7 days apart	10 IU or 20 IU or placebo; intranasal	45 min	Emotion recognition	Low dose increases true and false positives, high dose reduces false positives in polydipsic patients.	Emotion recognition improvement after 20 IU.
Woolley et al., 2014 [67]	RDBPC/CO	29 (29M) chronic SCHZ, 31 (31M) HC	2 sessions, approx. 1 week apart	40 IU or saline placebo; intranasal	30 min	Social cognition	Improved controlled (not automatic) social cognition.	Improved comprehension of indirectly expressed emotions, thoughts, and intentions.
Andari et al., 2021 [68]	RDBPC	20 (20M) SCHZ, 19 (19M) HC	1 session	24 IU or placebo; intranasal	45 min	Emotion recognition	Enhanced emotion recognition during emotion-based ball-tossing game.	Acute low-dose OXT modest effect on social cue processing.
Shin et al., 2015 [69]	RDBPC/CO	16 (16M) SCHZ, 16 (16M) HC	2 sessions, 1 week apart	40 IU or placebo; intranasal	45 min	Amygdala reactivity	Amygdala reactivity increased for happy and decreased for fearful faces.	Attenuated amygdala reactivity in SCHZ, but increased reactivity in HC.
Wigton et al., 2022 [70]	DBPC/CO	20 (20M) SCHZ or schizoaffective disorder	2 sessions, 1 week apart	40 IU or saline placebo; intranasal	45 min (fMRI testing), 90 min (fMRI task)	Social cognition, neural activity (fMRI)	Attenuated bias for happy faces and attenuated neural activity in right insula, bilateral temporal gyri, and amygdala.	Prosocial properties supported.
Pedersen et al., 2011 [71]	RDBPC	20 (17M, 3F) Paranoid or undifferentiated SCHZ	2 weeks	2 × 24 IU or placebo daily; intranasal	50 min after the AM dose on day 14	Clinical symptoms, social cognition	Improved identification of second-order false belief, clinical symptoms, suspiciousness, anxiety, and paranoia.	Therapeutic effect on social cognition supported.
Feifel et al., 2012 [72]	RDBPC/CO	15 (12M, 3F) SCHZ	6 weeks	2 × 20 IU daily for 1 week, 2 × 40 IU thereafter or placebo; intranasal	N/A	Cognition	Improved verbal memory: total recall, short delayed free recall, and total recall discrimination.	Supported therapeutic effect on cognition.
Halverson et al., 2019 [73]	RDBPC	68 (68M) SCHZ or schizoaffective disorder	12 weeks	2 × 24 IU or placebo; intranasal	N/A	Social cognition, empathy, social behavior	Improvement on Interpersonal Reactivity index, Perspective-Taking Subscale	Little evidence for therapeutic efficacy on social symptoms, empathy, and introspective accuracy.
Guastella et al., 2015 [74]	RDBPC/CO	22 (22M) SCHZ or schizoaffective disorder	2 sessions, 2 weeks apart	24 IU or placebo; intranasal	45 min	Social cognition, neurocognition	Improvement on the DANVA paralinguistic scale, and higher-order social cognition.	Acute therapeutic effects on higher-order social cognition.
Davis et al., 2014 [75]	RDBPC	27 (27M) SCHZ	4 sessions	40 IU or placebo; intranasal	30 min (2nd session)	Clinical symptoms, social cognition, neurocognition	Improved social cognition, facial emotion recognition, empathy, emotional intelligence.	Improved empathy with combination of OXT and social skills training.
Abu-Akel et al., 2015 [76]	RDBPC/CO	29 (19M, 10F) Healthy Participants	2 sessions, 7 days apart	24 IU or placebo; intranasal	45 min	Empathy	Enhanced empathy to pain from perspective of others.	Consideration of social salience hypothesis.
Buchanan et al., 2017 [77]	RDBPC	58 (47M, 11F) SCHZ or schizoaffective disorder	6 weeks	2 × 24 IU daily; intranasal	N/A	Negative symptoms, cognition	No group differences for cognitive and negative symptoms.	No evidence for OXT’s therapeutic efficacy.
Cacciotti-Saija et al., 2015 [78]	RDBPC	52 (36M, 16F) SCHZ	6 weeks	2 × 24 IU daily (+24 IU prior to each weekly session); intranasal	15 min	Social cognition and functioning, symptom severity	Association between spray usage and change in negative symptoms.	Reduced negative symptoms over time, no improvement in social cognition, functioning, and symptom severity.
Lee et al., 2019 [79]	RDBPC	28 (20M/8F) SCHZ or schizoaffective disorder	3 weeks	2 × 20 IU or placebo; intranasal	N/A	Social cognition, social functioning	No difference between treatment groups	No evidence for OXT’s therapeutic efficacy.
Horta de Macedo et al., 2014 [80]	RDBPC	20 (20M) SCHZ, 20 (20M) HC	2 sessions, 15 days apart	48 IU; intranasal	50 min	Emotion recognition	No effects on facial emotion recognition.	No evidence for therapeutic efficacy.
Bradley et al., 2021 [81]	RDBPC/CO	26 (25F) SCHZ, 38 (38F) HC	1 session	40 IU or placebo; intranasal	45 min	Mentalizing	No evidence of OXT’s effect on mentalizing	OXT treatment may have sex-specific effects

DB = double-blind; CO = cross-over; F = female; HC = healthy controls; IU = International Units; M = male; OXT = oxytocin; NR = non-randomized; OP = open-label; R = randomized; PC = placebo-controlled; SCHZ = subjects with schizophrenia. * Conclusions based on authors.

**Table 2 brainsci-13-00461-t002:** Preclinical Studies.

Author, Year	Animal Model	Trial Duration	OXT Administration	Outcome Measures	Results (OXT)	Conclusions *
Ferguson et al., 2002 [14]	Male OXT −/− and OXT +/+ (Hybrid mice constructed from 129S7/SvEvBrd-Hprt^b-m2 and C57BL/6J background strains, N = 42 per genotype)	4 administrations, 48–72 h apart	1 ng; intracerebroventricular	Social memory	OXT −/− mice no social memory, OXT +/+ mice intact social memory; OXT restored social memory, OXT antagonist produced social amnesia.	OXT essential in development of social memory in mice.
Feifel and Reza, 1999 [86]	Male Sprague-Dawley rats (N = 32); weight 225–250 g	4 sessions, 7 days apart	0.04–1 mg/kg; subcutaneous	sensorimotor gating (intact ppi and ppi disrupted by apomorphine, amphetamine, and dizocilpine).	OXT dose-dependently antagonizing effects of amphetamine and dizocilpine on PPI.	OXT receptor may modulate dopaminergic and glutamatergic regulation of PPI.
Tapias-Espinosa et al., 2021 [87]	Naïve male HS (“National Institutes of Health genetically heterogeneous” rat stock, N = 46), inbred Roman high-avoidance (N = 54) and Roman low-avoidance (N = 45) rats; 3–4 months; weight 320–390 g	HS rats: 2 sessions, 7 days apart; RHA, RLA rats: 1 session	0.04 mg/kg or 0.2 mg/kg; subcutaneous	Sensorimotor gating (PPI), oxytocinergic mechanisms (OXTR and CD38) in medial prefrontal cortex (mPFC).	Increased PPI in HS rats, attenuated PPI deficits in RHA rats, no effect in RLA rats; increased OXTR expression in RHA and RLA rats.	Antipsychotic-like effects likely related to OXT-related gene expression influences in mPFC.
Bowen et al., 2011 [88]	Male Australian Albino Wistar (AAW) rats (N = 48); PND 33 at dosing; weight 127–177 g	10 days (PND 33–42)	1 mg/kg; intraperitoneal	Physiological and behavioral effect during a key developmental epoch.	Reduced anxiety-like behavior (PND 50); Enhanced social interaction (PND 55).	Effect on sociability and anxiety reduction supported.
Kohli et al., 2019 [89]	Male Lister-hooded rats (N = 56); weight 150–200 g	4 sessions	0.03–0.3 mg/kg; subcutaneous	Locomotor activity, core body temperature, social behavior	Enhanced social interaction between unfamiliar rats, and nucleus accumbens dopamine release; attenuated hyperactivity	Therapeutic potential of oxytocin in social behavior.
Chang et al., 2012 [90]	Male rhesus macaques (Macaca mulatta) (N = 2)	12 OXT, 10 saline placebo sessions (alternating days)	25 IU; intranasal	Social cognition	Enhanced preference and attention to reward other monkey over time, enhanced prosocial choices.	Enhanced social donation behavior, other-oriented attention, decision times.
Rault et al., 2013 [91]	Male (castrated) and female pigs (progenies of Yorkshite x Landrace dams bred to Duroc sires, N = 43); 1–3 days of age	3 days	24 IU; intranasal	Social stress	Increased aggression in social mixing, greater cortisol concentrations.	Long-term dysregulation of HPA axis, increased aggression, and decreased social contact.
Huang et al., 2014 [92]	Male C57BL/6J mice; 12 and 20 weeks of age	7–21 days (chronic treatment); 1 session (acute treatment)	2 × 0.15 IU or 0.3 IU; intranasal	Social behavior	Acute treatment increased social behavior to opposite-sex unfamiliar subjects (vs. same-sex unfamiliar subjects).	Different social behavior effect of chronic vs. acute administration.
Bales et al., 2013 [93]	Prairie voles (*Microtus ochrogaster*) (N = 89)	21 days	0.08–8.0 IU; intranasal	Social behavior	Acute treatment enhanced social behavior; chronic treatment reduced partner preference behavior	Difference in long-term and short-term OXT treatment.
Parr et al., 2013 [94]	Rhesus macaques (4M, 2F);	1 session	48 IU; intranasal	Social perception	Reduced attention to negative stimuli.	Monkey social perception mediated by OXT.

IU = International Units; OXT = oxytocin; PPI = pre-pulse inhibition. * Conclusions based on authors.

## Data Availability

Data sharing not applicable.

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
