# Peer review of "Exploring a Possible Interplay between Schizophrenia, Oxytocin, and Estrogens: A Narrative Review"

_brainsci, 2023, doi:10.3390/brainsci13030461_

Round 1

Reviewer 1 Report

Thank you for the opportunity to review the manuscript.

Below, I post some considerations that I consider important for later publication:

Title:

I believe it is important to describe the type of study in the title, so that the reader has an idea of what the article is about.

Abstract:

In general, the abstract is expected to contain important information about all parts of the review. While the introduction is very long, I felt that the methodology, the results, and the conclusion of the study were missing.

I think it is important to report the type of review you have constructed, especially in the goal part. From how it was written, I believe it is a narrative review. But, this needs to be clear.

At one point in the abstract the authors report that there were 21 studies listed, soon after, that there were 23. How many were there? I suggest they correct this information.

This comment extends to the lack of a topic on the methodological aspects of the study.
How did you arrive at this N? Even without being a systematic or integrative review, I believe that providing some methodological information would make it easier to understand the research question. e.g., what were the eligibility criteria?
did you include only studies that evaluated women?
the objective refers to a search for current scientific evidence, so what is the time period in which the search was performed?
Moreover, even if it is a narrative review, checklists made to guide the construction of reviews, such as SANRA, suggest that the search strategy used should be described. For example: which descriptors were used? search period? eligibility criteria?
This is important because it makes the methodological path used clearer so that other researchers can redo the research.

In the objective still presented in the abstract, the authors point out: "a plausible interaction between hormonal variations of oxytocin and estrogens in female patients diagnosed with schizophrenia".
Was it only women? That needs to be clear in the eligibility criteria and in the title. Also, if it was only women, why the discussion about men during the text ?
What would be "a plausible interaction"? What do the authors expect from this interaction? What are the hypotheses of the study?
Information about the purpose and hypothesis is also needed at the end of the introduction.

Keywords: Wouldn't it be interesting to restrict the keywords to more specific aspects of the research? e.g. oxytocin, schizophrenia ...

Introduction:

The introduction has some inconsistencies.
I believe that the authors could better inform the importance of the study on a scientific and social level.
Why do the review? Aren't there reviews that seek to summarize the results of the literature?
If there are, what are the limitations that your review will seek to remedy?
Also, the last sentences of the last paragraph of the introduction are with aspects of conclusion, which makes the text a little confusing.
Ideally, the introduction should follow a funnel strategy, starting with more general aspects and ending with the research objective. I suggest that the authors can readjust these details.

The authors report in the introduction that studies evaluating the purpose of the study are scarce, are why the importance of the review? Shouldn't a review be used to summarize evidence? Why summarize something that is scarce?

In the last paragraph of the introduction, the authors report on hormonal variations and the menstrual cycle. For this aspect to have been cited even in the introduction, was it an eligibility criterion in the sample? In my perception, the research question is not entirely clear and could be elucidated better.

The topic 2: “The Oxytocin System and Schizophrenia”, It presents introductory ideas, the content does not portray the relationship presented in the title, because little addresses this relationship. on the contrary, it introduces in a very introductory way how the neurobiological mechanism that can cause the relationship and suggests that the effects of this interaction will be presented in the next topic.
It is important that scientific writing is done in an objective way, without redundancy. I suggest that the authors can try to make the text more synthesized and objective.
Throughout the text some points are a little confusing, I suggest that the authors could revise the writing and reorganize the ideas better. Also, some sentences are without references. I suggest that this should also be corrected.

In topic 2.1 the authors report on the Endogenous Oxytocin System in Schizophrenia.
The topic is subdivided into subtopics. It starts by talking about the increase or decrease of oxytocin.
Why might this occur? It is important that in addition to showing the contradictions, the authors can present why they appear. Why do some show an increase and others a decrease? What can the increase of oxytocin in patients with schizophrenia cause?
I have missed going more in-depth into the sentences that are presented. You address these topics in the later topics, but I believe it would be more interesting to talk about what the studies present and discuss this factor. In my opinion, this would help to make the reading more fluid, without the presence of cut ideas during the text.

I found that topics 2.2 and 2.3 have some problems.
The authors report an informative table about the studies found and the main information, and then report again the results of the studies in a very prolix way.
I believe that it is not necessary to report details of the studies individually again in the text, since these characteristics are presented in table form.
I suggest that the authors can evaluate the main ideas of the results found, and report back to the text in a summarized form.

I don't believe that figures 1 and 2 have much relation to the general objective of your study, so I don't feel the need to present them in this article, since they are information that could be described in the body of the text. We usually use figures to depict important aspects of our study.

I didn't understand the description of topic 5: Methodological Considerations.

From what is described, wouldn't these be the final considerations of the article?

Even if they are the final considerations, what are the limitations of this study? What are the perspectives of future articles to remedy these limitations?

Many of the references are very outdated.
Ideally, at least 75% of the references should be from the last five years. This may vary in reviews that are performed from the date of first publication, but the inclusion period of the articles is not expressed in the text, so we are unable to draw conclusions about this.
However, even in review articles, it is important that the references used in the introduction are up to date, because this provides the current overview of the phenomenon studied, and helps to explain the importance of the study.

In addition to the changes presented, I suggest that the authors restructure the manuscript considering the Scale for the Assessment of Narrative Review Articles - SANRA.

It is a great guide for writing narrative reviews, and can help in improving the final mauscrit.

Yours sincerely.

Reviewer 2 Report

This is a concise review on possible interplay between oxytocin and estrogens in female patients with schizophrenia.  Both pre-clinical and clinical evidence for OXT – ER interrelationship have been convincingly presented. Also therapeutic value of exogenous oxytocin, including its intranasal administration has been sufficiently discussed. Overall, this is a well written and quite interesting paper.

Specific remarks:

  1. A relevant paper by González-Rodríguez and Seeman  has not been cited (The association between hormones and antipsychotic use: a focus on postpartum and menopausal women. Ther Adv Psychopharmacol. 2019). In this paper  the authors stated that “When used with antipsychotics, hormones may also affect the metabolism and, hence, the brain level of specific antipsychotics. This makes treatment with antipsychotics plus hormones complicated”. In the reviewed  paper possible interactions of various  antipsychotics with oxytocin and estrogens should also be mentioned.
  2. Which combinations of antipsychotics and oxytocin/estrogen might be recommended  in treatment of drug-resistant schizophrenia?
  3. Typograhic error: Page 4, Exoogenous Oxytocin

Round 2

Reviewer 1 Report

The manuscript has been sufficiently improved to justify publication in Brain Sciences.
I am in favor of publication.